# The Impact of Climate Change on Immunity and Gut Microbiota in the Development of Disease

**DOI:** 10.3390/diseases12060118

**Published:** 2024-06-03

**Authors:** Pierluigi Rio, Mario Caldarelli, Antonio Gasbarrini, Giovanni Gambassi, Rossella Cianci

**Affiliations:** 1Department of Translational Medicine and Surgery, Catholic University of the Sacred Heart, 00168 Rome, Italy; pierluigi.rio18@gmail.com (P.R.); mario.caldarelli01@icatt.it (M.C.); antonio.gasbarrini@unicatt.it (A.G.); giovanni.gambassi@unicatt.it (G.G.); 2Fondazione Policlinico Universitario A. Gemelli, Istituto di Ricovero e Cura a Carattere Scientifico (IRCCS), 00168 Rome, Italy

**Keywords:** climate change, immune system, gut microbiota, inflammation

## Abstract

According to the definition provided by the United Nations, “climate change” describes the persistent alterations in temperatures and weather trends. These alterations may arise naturally, such as fluctuations in the solar cycle. Nonetheless, since the 19th century, human activities have emerged as the primary agent for climate change, primarily attributed to the combustion of fossil fuels such as coal, oil, and gas. Climate change can potentially influence the well-being, agricultural production, housing, safety, and employment opportunities for all individuals. The immune system is an important interface through which global climate change affects human health. Extreme heat, weather events and environmental pollutants could impair both innate and adaptive immune responses, promoting inflammation and genomic instability, and increasing the risk of autoimmune and chronic inflammatory diseases. Moreover, climate change has an impact on both soil and gut microbiome composition, which can further explain changes in human health outcomes. This narrative review aims to explore the influence of climate change on human health and disease, focusing specifically on its effects on the immune system and gut microbiota. Understanding how these factors contribute to the development of physical and mental illness may allow for the design of strategies aimed at reducing the negative impact of climate and pollution on human health.

## 1. Introduction

Climate change describes prolonged alterations in temperatures and weather patterns [1]. Since the 1800s, human activities, primarily linked to the combustion of fossil fuels such as coal, oil, and gas, have been the primary agent related to this phenomenon. The burning of fossil fuels produces greenhouse gases that can elevate temperatures by trapping the heat of the sun. The Earth’s surface temperature is approximately 1.1 °C warmer than in the late 1800s, before the industrial revolution, and warmer than at any point in the last 100,000 years [2]. The decade 2011–2020 holds the record as the warmest, with each of the previous four decades surpassing any preceding decade since 1850.

As the Earth functions as a complex interconnected system, changes on one side can trigger repercussions across all others. The results of climate change involve drought, severe wildfires, elevated sea levels, floods, polar ice melting, dangerous storms, and reduced biodiversity [1]. The weather phenomena can directly and indirectly impact health by promoting chronic illnesses, infectious diseases, and health crises [3]. Heatwaves or hurricanes can directly cause injury, illness, and even fatalities [4]. However, several effects are indirect and involve shifts in the environment, which can subsequently affect human health. For instance, alterations in temperatures and rainfall patterns can influence the lifecycles of animals that can spread diseases, like Lyme or West Nile virus, potentially leading to new outbreaks [5]. Additionally, rising sea levels can exacerbate flooding from hurricanes in coastal regions, with exposition to contaminated water, pollutants, and hazardous waste. Climate fluctuations often coincide with other health stressors, such as poverty, social disparities, and impaired access to resources, collectively heightening vulnerability [4].

As estimated by the World Health Organization (WHO), in 2023, “one in four deaths can be attributed to preventable environmental causes and climate change is exacerbating these risks” [6]. Skevaki et al. have recently listed several climate-related exposures, including unfavorable weather events, extreme heat, air pollution, allergens, water and food insecurity, and changes in the distribution of pathogens and vectors, connecting them to climate-sensitive health outcomes such as cardiovascular, respiratory and infectious diseases, cancer, malnutrition, and mental illness [7].

Ray and Ming observed that climate change dramatically modifies antigen exposure and impairs antigen-specific tolerance, thus contributing to the onset of allergic and autoimmune diseases and the exacerbation of pre-existing immunologic disorders [4]. Climate-related stressors have proved to impair the efficacy of the epithelial barriers, which are the first line of defense against invaders, hyperstimulate the innate immunity and affect the adaptive one [7].

Moreover, there is growing evidence in the literature suggesting a crucial role of climate change and environmental pollutants in reducing both macro- and microdiversity. According to the ‘biodiversity hypothesis’, a biologically poorer natural environment negatively impacts on human microbiome composition and activity [8].

Several studies suggest the role of climate change in modifying the soil microbiome, which is inextricably linked to the quality of food crops [9]. Modern research has been delineating the influence of the soil microbiome on various physiological processes, such as digestion, vitamin and mineral production, mental health, and immune function [10]. Climate-related changes in crop quality may alter the human gut microbiota (GM) composition, for example, decreasing Bacteroides and increasing Proteobacteria, as typically observed in malnutrition [11].

It is well known that the imbalance of the microbiome, defined as ‘dysbiosis’, underlies several pathological conditions, including infectious, inflammatory, and autoimmune diseases [12].

In a period of increasing awareness about the ‘health emergency’ of climate change, this narrative review aims to explore the pathophysiological influence of climate change on human health, focusing specifically on its effects on immune responses and GM, and describing how they contribute to the development of disease.

Advancing current knowledge of the real impact of climate change on human health and disease may allow radical climate interventions, in order to shape a “liveable future” for humanity [13].

## 2. Climate Change in Health and Disease

In recent years, a growing interest in the health consequences of climate change has emerged. Several studies have evaluated the influence of this change on both physical and psychological health, as well as its related social and economic issues, mostly reported in underdeveloped countries [14].

In an overview of systematic reviews about the relationship between climate change and health, Rocque et al. came to four principal findings: the attention of research on meteorological factors, a prominent focus on physical health, the existence of fewer studies from low- or middle-income countries, and a general association between climate change and disease [15].

In this context, the main drivers of disease are the emissions of greenhouse gases, such as methane, nitrous and carbon oxide, and chlorofluorocarbons from human activities, the increased temperature worldwide, weather phenomena such as drought and storms, and the spread of allergens and infectious agents. The most investigated consequences on health from climate change concern the respiratory, cardiovascular, and digestive systems, together with a higher risk of infectious diseases (e.g., malaria, typhus, and cholera) and cancer. It has been supposed that global warming could mitigate the negative impact of lower temperatures on health, preventing the onset of cold-related illnesses (e.g., bronchopneumonia or articular disorders). However, the risks undoubtedly overcome the potential benefits [16].

For instance, in a hot and dry climate, the quality of air worsens due to air pollutants, leading to the exacerbation of chronic respiratory disorders and allergic diseases, particularly in vulnerable individuals, such as the elderly and children. Moreover, respiratory health could be damaged by the emergence and transmission of viral infections [17]. Andersen et al. evaluated not only the direct effects of climate emergency on patients with respiratory disorders, but also indirect effects, including social conflicts, economic issues, and the disruption of healthcare infrastructures that further increase the global burden of respiratory diseases [18]. It is known that air pollution is the second highest cause of lung cancer, as well as a negative factor in lung cancer survival. As highlighted by Berg et al., lung cancer mortality has increased by almost 30% in the last two decades, coinciding with a reduced smoking prevalence and a higher exposure to air pollutants. Indeed, outdoor air pollutants, including particulate matter 2.5 μm or less in diameter (PM_2.5_), have been defined as carcinogenic to humans by the International Agency for Research on Cancer [19]. Guo et al. found a positive association between PM_2.5_ air concentration and lung cancer incidence in China, showing a major impact of PM_2.5_ on health in regions with extreme climate conditions [20].

Climate change may lead to cardiovascular diseases [21]. Exposure to PM has been associated with the development of coronary heart disease, stroke, and arterial hypertension [22]. Additionally, Li et al. observed an increased number of emergency department visits due to ischemic heart disease, hypertension, and arrhythmias on extremely hot days, suggesting the impact of very high temperatures on cardiovascular risk [23].

Furthermore, a bidirectional interaction between climate change and physical activity (PA) can be identified. On one hand, pollution, extreme heat, and weather phenomena induce PA reduction, especially in the elderly, overweight individuals, and patients with chronic diseases; on the other hand, PA and sports both play a mitigating and amplification role (e.g., greenhouse gas emission caused by vehicles) in climate change [24].

Moreover, global warming and climate change may be responsible for the development and progression of gastrointestinal diseases. Several studies have highlighted an increased number of gastrointestinal infections, hepatitis, and cancer in susceptible individuals (e.g., patients with chronic illness) exposed to heat waves, air pollutants, drought, and flooding [25,26,27]. It has been reported that extremely high temperatures could induce human liver ischemia and hepatocyte necrosis, resulting from peripheral vasodilation and splanchnic vasoconstriction. At the same time, extreme heat may lead to direct liver and pancreatic cytotoxicity, and damage the gut epithelial barrier, increasing the intestinal permeability and the risk of endotoxemia [28]. The negative impact of various pollutants on gastrointestinal health has been evaluated. For instance, exposure to pesticides and chemical substances, used in response to crop failure related to the climate emergency, could lead to gut dysbiosis, which takes part in the pathogenesis of intestinal inflammatory disorders and colorectal cancer. Similarly, air pollutants, such as PM, have proved to induce inflammation and oxidative stress in different organs, including the gastrointestinal tract [29].

Interestingly, climate change is related to food in a bidirectional manner. On the one hand, climate influences food systems in different ways (e.g., changes in soil fertility and crop production due to unfavorable weather events, reduced water and food availability, changes in food market and prices, lower food nutrient concentration, and increased infections) [30,31]. On the other hand, various steps in the food production chain may impact on the environment by increasing the greenhouse gases emissions and rising temperatures [32]. A healthy balanced diet plays a pivotal role in preserving gut health and preventing the onset of cardiovascular, metabolic, infectious, and inflammatory disorders. Polyphenols and probiotics, typically found in vegetables, fruits, legumes, and seeds, have a positive impact on GM, increasing health-beneficial bacteria such as Bifidobacteria and *Lactobacillus*, which reduce inflammation and improve immune function [33]. For instance, *Lactiplantibacillus plantarum YW11*, belonging to the genus *Lactobacillus*, is involved in various metabolic pathways (e.g., carbohydrate metabolism) and protects against gut pathogens such as *Campylobacter*, *Yersinia*, *Shigella*, and *Salmonella* [34]. On the contrary, unhealthy diets high in sugars and saturated fats, and low in fiber (e.g., Western pattern diet) lead to an overgrowth of pro-inflammatory gut bacteria, an impairment of immune responses, and the development of both gastrointestinal and systemic inflammatory diseases [35].

Van de Vuurst et al. have recently investigated the leading trends of research on climate change and infectious diseases. Overall, they found a predominant number of studies regarding vector-borne diseases conducted in high-income temperate countries [36]. For this reason, the crucial role played by climate change in directly transmitted infections in tropical countries could be underestimated. Moreover, according to the authors, the anthropocentric perspective of research may limit the awareness of climate change’s impact on wildlife [36]. As observed by Mora et al., climate-related hazards aggravate over half of infectious diseases impacting humans worldwide. The authors found that warming and changes in rainfall patterns lead to a broader distribution of vectors (e.g., mosquitoes, ticks, and birds) implicated in various infectious diseases such as malaria, Lyme disease, and West Nile virus. Additionally, climate change has been promoting waterborne diseases (e.g., *Vibrio* bacteria or amoebic infections) due to increased water-related activities in high-temperature conditions. Furthermore, weather phenomena, such as extreme rains, could encourage human social gathering and the consequent transmission of airborne pathogens (e.g., severe acute respiratory syndrome coronavirus 2, SARS-CoV-2) [37].

In the last few decades, cancer has progressively become one of the most important causes of early death worldwide. As suggested by Yu et al., climate change can contribute to the onset and progression of cancer due to its interaction with several risk factors, such as ultraviolet radiation (UV), extreme temperature, air pollutants, food and water contaminants, and infections [38]. In addition, climate disasters could negatively influence global oncology, hindering patients’ access to cancer care [39].

Overall, climate change increases exposure to carcinogens. The heavy rain from Hurricane Harvey in Texas affected many factories, causing worries about oil and chemical spills [40]. It has been observed that exposure to PM_2.5_ increases the risk of epidermal growth factor receptor-driven lung cancer [41]. Similarly, wildland firefighters exposed to wood smoke have a higher risk of lung cancer [42]. Parker found an association between climate change and skin cancer incidence, mediated by different environmental factors, such as UV, high temperature, stratospheric ozone depletion, and pollutants [43]. Moreover, climate change could influence the presence of mycotoxins in food (e.g., crops). Among them, aflatoxin has shown potent carcinogenic activity [44,45].

Interestingly, a bidirectional relationship between climate change and cancer exists. With the term “climate toxicity”, Weadick et al. refer to the negative impact of healthcare workers and facilities on the environment, such as carbon production and greenhouse emissions due to both diagnostic and therapeutic interventions [46].

A mention should be made of the relationship between climate change and mental health. As observed by Walinski et al., extreme weather events (e.g., floods and hurricanes), chronic stressors (e.g., drought, extreme heat, fear of food contamination), and migration related to the climate emergency increase the risk of psychiatric illness, such as post-traumatic stress disorder, affective disorders and anxiety [47]. Climate change could play a disruptive role in conditions associated with well-being, such as environmental, social, economic, and cultural conditions, thus increasing both the collective and individual experience of insecurity and mental illness [48].

As discussed above, climate change has deleterious effects on both physical and mental health, particularly in vulnerable groups, such as patients with chronic diseases, children, and people living in low-income countries, who should be the target of appropriate strategies aimed at reducing the health burden of climate change (Table 1).

## 3. Climate Change and Sex Differences

As indicated by the United Nations, “the climate crisis is not gender neutral” since women and girls experience its worst consequences in terms of wellbeing. Across the world, climate change may amplify political and social tensions, leading to conflicts in which women become victims of gender-based violence. In the context of climate calamity, the disparities between men and women in access to resources (e.g., health care) increase, thus rendering women more vulnerable and susceptible to illness or death. In addition, during periods of extreme rainfall or drought, activities generally involving women in low-income countries, such as agriculture, may be significantly impaired [49]. Furthermore, due to intrahousehold food intake inequalities, women and girls often experience food insecurity [50].

Climate change also directly influences women’s health, in particular during menstruation and pregnancy, which are periods with specific nutritional and physiological needs [51]. Maternal health is closely related to pediatric health and consequently overall population health.

Women experience crucial immunological and hormonal changes aimed at achieving a successful pregnancy [52,53]. During pregnancy, women are particularly sensitive to environmental temperature. In a retrospective cohort study, Jiao et al. found that both long- and short-term maternal heat exposure increases the risk of severe maternal morbidity [54]. Among the physiological adaptations related to pregnancy, thermoregulation plays a pivotal role. Inappropriate heat dissipation could lead to negative maternal and neonatal outcomes, such as miscarriage, preterm birth, low birth weight, and congenital disorders. Wyrwoll described several biological mechanisms potentially underlying these pregnancy complications, including an impaired placental function, modifications in uterine contractility, dehydration, and inflammation [55]. In this context, emerging evidence supports the role of the endocrine system, particularly the hormonal imbalance caused by climate change, in threatening reproductive success. Extremely high temperature has been associated with lower ovarian estrogens and progesterone, increased cortisol levels, and reduced activity of the placental enzyme 11-beta-hydroxysteroid dehydrogenase 2, normally converting cortisol into an inactive form. As a result, after crossing the feto-maternal barrier, cortisol influences fetal development. Moreover, in response to heat exposure, reduced concentrations of progesterone, a hormone with anti-inflammatory effects involved in pregnancy immune tolerance, may promote inflammation and fetal loss [56].

Furthermore, Van Daalen et al. observed that women and children have generally a higher exposure to indoor PM, derived from cooking and heating stoves, compared to men. In many parts of the world, according to traditional gender roles, women dedicate more time to indoor activities, thus inhaling greater amounts of air pollutants. Among the consequences to health, reproductive, respiratory, and cardiovascular disorders have been reported [50,57].

Sex differences may exist also in the susceptibility to disease following exposure to outdoor air pollutants. For instance, Liu et al. detected an association between ozone (O_3_) exposure and cerebrovascular disease mortality, and women appeared more vulnerable than men [58]. Moreover, Kim et al. found a higher risk of reduced cognitive function after exposure to PM and nitrogen dioxide (NO_2_) in women compared to men [59]. However, a Polish study evaluating the impact of air pollutants on mortality found greater cardiovascular mortality in men following exposure to PM_2.5_ compared to women [60].

Despite the existence of sex differences related to climate change, sex-related perspectives are often poorly represented in both medical and environmental research. Additionally, women take part more rarely in the climate change debate and adaptation policies, especially in developing countries [50]. One of the reasons could be the lack of informational campaigns. A recent study conducted in Turkey demonstrated a direct association between women’s climate change awareness and their concerns about it [61], which allows for the design of strategies to face climate emergencies.

## 4. Climate Change and the Immune System

The immune system encompasses both innate and adaptive mechanisms aimed at defending the host against microbial pathogens, toxicants, or allergens, through an accurate discrimination of the self from the non-self [62].

In the last decades, it has been clarified that environmental factors, such as diet, lifestyle, and pollution, can affect immune function and contribute to the development of autoimmune and allergic diseases [63]. In this context, according to Swaminathan et al., the immune system represents an important interface through which global climate change impacts human health. As suggested by the authors, undernutrition, UV radiation exposure, and psychological stress could be a few of the mediators of climate-related immune imbalance, resulting in an increased risk of infections, and allergic and autoimmune diseases, particularly in vulnerable individuals such as children and the elderly [64].

### 4.1. Global Warming

Global warming is one of the crucial issues of climate change. Since the second half of the twentieth century, an increased occurrence of heat waves has been reported worldwide, constituting significant weather-related harm to human health [65,66]. Global warming is associated with an increased risk of infectious diarrhea, respiratory diseases, and epidemics, with strong pressure on the immune system to preserve homeostasis. Epidemics could be determined by the emergence of novel pathogens, adapted to higher temperatures, and the wider distribution of disease vectors [67].

Environmental temperature influences the body temperature of animals. In a very high-temperature environment, animals develop physiological responses to heat, collectively defined as ‘heat stress’. Heat stress leads to the neuroendocrine activation of the hypothalamus–pituitary–adrenal axis with the consequent production of glucocorticoids, which increase the glucose levels under stress conditions, but may also cause cardiovascular and immune impairment. In addition, heat stress has proved to disrupt the balance between anti- and pro-inflammatory cytokines, causes oxidative stress, and damages the tight junctions, increasing intestinal permeability [68].

Under normal conditions, the human body temperature fluctuates around the mean depending on sex, age, and circadian rhythm. The existence of a link between body temperature and immune response has been identified, contributing in some cases to the disease processes. For instance, in rheumatoid arthritis, higher temperatures in joints promote the activation of T helper 17 (Th17) cells, which in turn stimulate the release of inflammatory cytokines and heat shock proteins (HSPs), enhancing the inflammatory response and heat production [67]. Since hyperthermia and oxidative stress can disrupt important cell functions such as protein folding, HSPs represent a protective response, allowing for protein stabilization and trafficking, preventing protein misfolding, and reducing the reactive oxygen species (ROS) levels through glutathione production [69].

Presbitero et al. evaluated the effects of heat stress on the human innate immune system. They found that a mild increase in core temperature positively impacts innate immune response, whereas core temperatures above 38 °C impair the efficacy of immune response and exacerbate inflammation [70].

Several animal studies investigated the effects of climate change on immune function. For instance, the increased seawater temperature causes tissue damage and reduced phagocytic response against the bacterium *Vibrio parahaemolyticus* in the sea anemone *Exaiptasia pallida* [71]. Similarly, marine mussels are more vulnerable to the pathogenic bacterium *Photobacterium swings* at high temperatures [72]. Folkertsma et al. demonstrated that temperature is a crucial determinant of genomic variation in bank voles, aimed at adapting to the environmental conditions. Interestingly, the authors identified loci related to climate adaptation within genes involved in immune responses and lipid metabolism [73]. Kim et al., analyzing the stress response of dairy cows to high temperatures, found significant changes in energy metabolism and immune function, with a general promotion of inflammation and mitochondrial activity. Among the differentially expressed genes (DEGs), under heat stress conditions, the authors detected an upregulation of genes related to the interleukin (IL)-17 pathway, oxidative phosphorylation, and antimicrobial humoral responses, and a downregulation of genes involved in collagen organization [74]. As indicated by Martin et al., rising temperature encourages immune senescence in the mosquito *Anopheles gambiae*, a vector of malaria. In particular, at higher temperatures, the researchers observed the weakening of melanization, a humoral immune response against malaria parasites, and changes in gene expression related to immune function (e.g., Toll pathway and apoptosis) which facilitate the transmission of the disease [75].

It has been established that heat stress could contribute to genomic instability through both DNA strand breaks and oxidative damage. Moreover, heat stress disrupts DNA repair mechanisms and cell death pathways, thus promoting the survival and proliferation of DNA-damaged cells. In this context, heat stress acts as a carcinogen. Furthermore, people with an occupational risk of heat stress are also more frequently exposed to known carcinogens, such as polycyclic aromatic hydrocarbons (PAHs) or pesticides. These workers are likely to develop negative health consequences due to increased skin permeability and respiratory rate in response to heat exposure and the consequent thermoregulation [76]. In a Spanish case–control study, Hinchliffe et al. detected a significant association between occupational heat exposure and the risk of breast cancer in women [77]. The link between heat stress and the impairment of cellular responses to UV-induced DNA damage explains skin carcinogenesis [78].

In the last few years, an area of growing interest is represented by the relationship between climate change and allergic disorders. The rising environmental temperatures and the novel meteorological patterns prolong the pollen season, thus increasing the allergen exposure of sensitive individuals and exacerbating allergies [79]. Moreover, the concomitant exposure to air pollutants, such as PM, O_3_, NO_2_, and sulfur dioxide (SO_2_), worsens the severity of symptoms in pollen-allergic populations [80]. As observed by Singh and Kumar, the increased atmospheric levels of greenhouse gases, such as carbon dioxide (CO_2_), which is a consequence of the burning of fossil fuels, can promote plant growth, pollen dispersal, and transport, and ultimately the appearance of new allergens in non-endemic areas [81].

Epigenetic mechanisms could contribute to the development of allergic and auto-inflammatory disorders in response to elevated temperatures, pollen, and pollutants. For instance, at a pulmonary level, PAHs, derived from the combustion of organic materials, activate the aryl hydrocarbon receptors (AhRs), which influence gene expression through xenobiotic response elements (XREs). This process leads to changes in the methylation of genes encoding immune molecules, such as IL-4, IL-5, IL-9, IL-13, GATA-binding protein 3 (GATA-3), and T helper 2 (Th2) cytokines, or related to regulatory T cell (Tregs) function [82]. Similarly, in response to pollen exposure, naïve T-helper cells differentiate into Th2 cells that in turn activate other immune cells, such as B cells and eosinophils. In this context, epigenetic regulation takes part in maintaining the Th2 phenotype and promoting inflammation [82]. Epigenetic mechanisms are an interface between the environment and the genome. Since they take part in crucial cellular functions (e.g., cell cycle, gene expression, and DNA repair), altered epigenetic events could play a role in the development and progression of cancer [83].

Çelebi Sözener et al. investigated the effects of climate change on the respiratory epithelial barrier. They found that extremely high temperatures, allergenic pollens, and pollutants disrupt the tight junctions of respiratory epithelium. The authors observed several direct impacts of extreme heat on the respiratory system, including a decrease in ciliary beat, disrupted mucus production, heightened airway reactivity, and an increased inflammatory response, indicated by higher levels of tumor necrosis factor-α (TNF-α), IL-1β, IL-4, and IL-6. They also noted oxidative stress, activation of transient receptor potentials vanilloid 1 and 4, and the induction of HSPs. Additionally, global warming indirectly heightens the risk of asthma and allergic rhinitis through various climatic factors. Increased humidity, allergens, sand and dust storms, air pollution, and wildfires have been linked to more significant epithelial barrier damage and greater permeability [84] (Figure 1).

### 4.2. Fires, Storms, and Pollutants

A mention should be made of wildfires, which have occurred more frequently in recent years due to global warming. As indicated by Albery et al., wildfires change the patterns of wildlife disease since they weaken the animals’ immune responses and encourage the onset and spread of infections [85]. Wildfires lead to the environmental release of PM, carbon monoxide (CO), CO_2_, hydrocarbons, and heavy metals, which may harm respiratory health. Indeed, several studies reported a higher incidence of chronic obstructive pulmonary disease (COPD) exacerbations and asthma following exposure to wildfires [82]. Wildfires have proved to damage the human airway epithelium, altering DNA methylation and impairing the activation of T helper 1 (Th1) and Th2 cells [86]. In a child population study, Prunicki et al. described an association between wildfires and the increased Forkhead box P3 (Foxp3) methylation, together with a reduction in Th1 cells [87].

Furthermore, other environmental phenomena, such as sandstorms and desert dust, have shown negative effects on human health, particularly cardiovascular and respiratory functions [88,89]. As reported by UN experts in 2023, 2 billion tons of dust and sand enter the atmosphere each year (approximately 25% related to human activities) [90]. Sand and dust storms are not only a source of PM, O_3_, and CO_2_, but also a vehicle for microorganisms (e.g., bacteria, fungi, and spores) [91]. Once inhaled, dust particles activate airway dendritic cells, macrophages, and innate lymphoid cells, causing inflammation. The interaction between APCs and T helper cells leads to adaptive immune responses, including the activation of T cytotoxic cells and B cells, as well as the release of chemokines and cytokines that play a role in several diseases, such as asthma, COPD, silicosis and pulmonary arterial hypertension [92]. Bredeck et al. demonstrated that Saharan dust induces oxidative stress in lung epithelial cells and IL-1β release. However, they found that the latter was lower when microbial components were eliminated through heat, thus suggesting the pivotal role of microbial constituents (e.g., endotoxin) in desert dust toxicity. Furthermore, the activation of macrophage NLRP3 (nucleotide-binding domain, leucine-rich-containing family, pyrin domain-containing-3) inflammasome was observed after dust exposure [93].

Overall, the exposure to environmental pollutants has proved to increase cancer risk. Researchers found an association between air pollutants, such as PM and NO_2_, and respiratory, gastrointestinal, and reproductive cancers [94]. Inflammatory pathways (e.g., nuclear factor kappa B, NF-kB), oxidative stress, epigenetic mechanisms (e.g., DNA methylation and histone modification), and changes in microRNA seem to be the main drivers of pollution-related carcinogenesis [95,96]. For instance, in a mouse model, Hill et al. found that PM promoted the malignant transformation of EGFR-mutant alveolar type II cells in lungs, through the release of IL-1β by macrophages [41].

As mentioned above, climate change is accompanied by growing environmental concentrations of ozone, mainly resulting from anthropogenic activities [97]. Whereas stratospheric O_3_ is extremely positive for human health since it absorbs harmful UV radiation, surface-level O_3_ can cause damage to plants, animals, and humans, so it is also called “bad” ozone. O_3_ is a highly oxidizing gas, responsible for oxidative stress and inflammation in the lungs. O_3_ exposure has proved to increase the alveolar macrophage production of TNF-α, IL-lβ, IL-6, and IL-8, and the airway epithelial cell production of fibrinogenic proteins [98]. Moreover, acute O_3_-induced neutrophilic inflammation and the activation of NF-kB-dependent pathways have been described in airways [99]. Beyond the respiratory tract, O_3_ exposure has been associated with systemic inflammation, which plays a role in the development of neurological and cardiovascular diseases [100,101].

## 5. Climate Change and the Gut Microbiome

Climate change is forecasted to alter the composition of soil microbes, as temperature is one of the factors influencing the growth rate of organisms. With a decrease in the number of microbes in the soil, there could be a potential depletion of the GM in humans due to a reduction in soil biodiversity [9]. The low organic content of the soil will also affect the quality of crops and agricultural productivity. This could result in a depletion of micronutrients in crops that are essential for their fundamental effects on the human GM (by acting as co-factors in metabolism, regulating enzyme activities, or serving as coenzymes) [102]. An alteration in the quality of food crops could lead to changes in the composition of the intestinal microbiota, as indicated by a higher ratio of Bacteroidetes to Firmicutes, along with enhanced immunity in mice models [103].

The Firmicutes-to-Bacteroidetes ratio remains a topic of interest. Despite numerous studies analyzing this ratio, conflicting results have been reported. An increased abundance of Firmicutes has been linked to improved health outcomes in domains related to physical capabilities and cognitive function well-being [104].

The Firmicutes-to-Bacteroidetes ratio is often cited as an indicator of obesity. It has been hypothesized that discrepancies in the validity of this ratio may be attributed to interpretative bias derived from methodological differences, inadequate characterization of recruited subjects, and overlooking lifestyle-related factors known to influence microbiota composition [105].

Risely et al. analyzed 1141 fecal samples collected between 1997 and 2019 to investigate trends in β-diversity throughout the study [106]. They observed a significant shift in microbiota composition, characterized by increases in *Bacteroides* and *Fusobacterium*, and decreases in lactic acid bacteria (Figure 2).

### 5.1. Heat

Heat-related illnesses can result in morbidity, and it is anticipated that their frequency will increase with predictions of rising global surface temperatures and more frequent extreme weather events.

Liu et al. recruited 32 healthy young soldiers and divided them randomly into four teams to undergo heat acclimation training (HAT) for 10 days: the Equipment-Assisted Training team in high temperature (HE), the Equipment-Assisted Training team in normal hot weather (NE), the High-Intensity Interval Training team in high temperature (HIIT), and the Control team without training [107]. They discovered that the effects of HAT on participants in the HE team surpassed those in the NE team. Within the Heat Acclimation (HA) group, the differences in physiological indicators and plasma organ damage biomarkers (ALT, ALP, creatinine, LDH, α-HBDH, and cholinesterase) before and after a standard heat tolerance test were notably reduced compared to the test performed before the study. However, the differences in immune factors (IL-10, IL-6, CXCL2, CCL4, CCL5, and CCL11) increased. There were significant changes in the composition of GM, characterized by a decrease in the proportion of potentially pathogenic bacteria (*Escherichia*, *Shigella*, and *Lactococcus*) and an increase in probiotics (*Dorea*, *Blautia*, and *Lactobacillus*) within the HA group [107] (Table 2).

### 5.2. Cold

Recent research has revealed that exposure to cold triggers alterations in the diversity and quality of GM species [110]. Short-term or sporadic exposure to cold did not appear to result in a general reduction in the α-diversity of GM. Yi et al. showed that subjecting organisms to cold stress (6 °C for 7 days) notably heightened the diversity of GM, a phenomenon also observed experimentally in piglets exposed to 18 °C for 48 h [112]. Nevertheless, prolonged exposure to cold typically leads to reduced richness and decreased α-diversity. Wang and colleagues examined the GM of rats following 6 weeks of cold exposure at 4 °C and observed that the α-diversity, measured by Chao1 richness and Shannon diversity indexes, was significantly diminished compared to the control groups [108].

Exposure to cold temperatures has the potential to harm the morphology and structural integrity of the colon, leading to the disruption of tight junctions within the colonic epithelial tissue. This disruption can result in increased gut permeability, apoptosis (cell death), inflammation, and oxidative stress. Additionally, exposure to cold can trigger the secretion of pro-inflammatory cytokines, such as interleukin-1β (IL-1β), interleukin-6 (IL-6), and TNF-α, further exacerbating inflammatory responses within the colon [109].

Cold signals can be detected by a cluster of TRP (transient receptor potential) channels on the skin, which are then relayed to the preoptic area in the brain. These TRP channels can also be activated by lipopolysaccharides (LPSs) secreted by the GM. The activation of TRP channels can facilitate the sympathetic nervous system to release β-adrenergic agonists, such as norepinephrine. This release is mediated by the stimulation of β3-adrenergic receptors (β3-AR) and subsequent activation of the protein kinase A (PKA) signaling pathway [113].

The β-adrenergic agonists can bind to β-adrenergic receptors on brown adipose tissue (BAT), initiating a series of downstream reactions. This cascade includes the activation of uncoupling protein 1 (UCP1) [114,115].

Furthermore, cold-induced changes in the GM and their metabolites, such as short-chain fatty acids (SCFAs) and secondary bile acids, can promote adaptive thermogenesis in BAT through their receptors. Specifically, SCFAs, particularly butyric acid, and their receptors can activate the cAMP-PKA-pCREB signaling pathway, leading to the secretion and release of NE in BAT. NE manipulation can also induce alterations in GM composition [110].

Moreover, cold temperatures can stimulate the thyroid gland to release triiodothyronine (T3), which plays a role in modulating UCP1 expression in BAT to further promote thermogenesis. The hypothalamic–pituitary–adrenal axis regulates and controls thyroid hormones through a negative feedback mechanism. SCFAs and the LPS can modulate local levels of T3 by enhancing the conversion of thyroxine (T4) to the biologically active T3 in the gut and liver [111] (Table 2).

Separately from its direct effects on metabolism and thermogenesis, thyroid function is closely connected with the GM. For example, patients with hyperthyroidism exhibit an elevated abundance of pathogenic GM, such as *Clostridium* and Enterobacteriaceae [116].

### 5.3. Pollution

Climate change is also linked to the rise in PM_2.5_, which has emerged as a significant worldwide issue for public health [117]. Li et al. additionally demonstrated that mice exposed to ultrafine particles exhibited elevated levels of Verrocomicrobia, but decreased levels of the phyla Cyanobacteria, Actinobacteria, and Firmicutes [118]. Liu et al. found that mice exposed to PM_2.5_ showed elevated proportions of the phyla Candidatus, Saccharibacteria, Proteobacteria, and Fusobacteria, while experiencing reduced proportions of the phyla Gemmatimonadetes, Acidobacteria, and Deferribacteres in their gut [119].

In a real-world perspective repeated panel study conducted in Jinan, China, significant alterations were discovered in the GM linked to exposure to PM_2.5_ resulting in alterations of tryptophan metabolism [111].

Therefore, it is evident that climate change and its associated consequences have implications for the gut–brain axis. White et al. conducted an integrative review, identifying 104 significant articles shedding light on the mental health effects of climate change on vulnerable populations worldwide. Their findings indicate that individuals vulnerable due to their geographical location, Indigenous communities, children, elderly individuals, and climate migrants experience a disproportionate impact from climate change-induced mental health issues. These include thoughts of suicide, depression, anxiety or eco-anxiety, post-traumatic stress disorder, sleep disturbances, substance abuse, and behavioral disruptions [120].

Furthermore, the term “Solastalgia” represents a modern concept aimed at comprehending the connections between human wellbeing and ecosystem health. It focuses on the combined effects of climate and environmental changes on mental, emotional, and spiritual health [121].

A significant correlation between mental health and functional gastrointestinal (GI) disorders, which are the most common GI disorders observed in the general population, exists [122]. Acute dysbiosis may manifest with mild abdominal discomfort and diarrhea. Climate change’s impact on mental health could potentially elevate the prevalence of functional GI disorders, such as irritable bowel syndrome (IBS). Additionally, exposure to certain micropollutants may contribute to an increased incidence of IBS [25]. However, it is crucial to note that chronic dysbiosis could elevate the likelihood of developing inflammatory bowel disease (IBD) [123].

Moon and colleagues conducted a meta-analysis on 7346 publications and concluded that a noteworthy connection existed between exacerbations of IBD and seasonal fluctuations [124].

They showed that within the ulcerative colitis subgroup, there was a noticeable but weak correlation with seasonal variation, while within the Crohn’s disease subgroup, there was a weak positive correlation observed with exacerbations. Additionally, the studies indicated a notable positive correlation between air pollutants and exacerbations of IBD [124].

Notably, dysbiosis and inappropriate levels of microbial byproducts in the intestinal tract could lead to the development of colorectal cancer.

Recent evidence suggests that certain bacteria, including *Fusobacterium nucleatum*, *Streptococcus bovis*, *Helicobacter pylori*, *Bacteroides fragilis*, and *Clostridium septicum*, play a role in the development of colorectal cancer (CRC) [125]. Metagenomic studies have confirmed that these bacteria are more abundant in CRC patients compared to healthy individuals. They can induce inflammation and DNA damage, contributing to cancer development. Moreover, these bacteria produce metabolites such as secondary bile salts derived from primary bile salts, hydrogen sulfide, and TMAO, which likely promote inflammation and subsequently cancer development [125].

## 6. Conclusions

Climate change is not just global warming: it is a multidimensional problem. As a result, environmental modification leads to changes in the soil microbiota, which in turn impact the foods we consume, causing alterations in the GM [126]. “We are what we eat”, said the German philosopher Ludwig Feuerbach. Furthermore, climate change and the alterations to the GM that it determines are intricately linked to the activation of innate immune responses. Various impacts of climate change, both direct and indirect, elicit the activation of innate immune cells, such as neutrophils, eosinophils, monocytes, basophils, and mast cells. Numerous triggers, both particulate and nonparticulate, activate similar pathways and effector responses, often synergizing with NF-κB-dependent pathways and stimulating the NLRP3 inflammasome, thereby eliciting acute inflammatory responses [7].

Moreover, the breakdown or absence of immune tolerance can initiate a broad range of noncommunicable diseases, including autoimmune disorders, allergies, respiratory and metabolic disorders, obesity, and various other clinical conditions [4].

As new evidence emerges, further research is needed to clarify the complex interplay between climate change and immune response, encompassing several global contexts by employing modern biological and epidemiological methodologies.

In the future, understanding the real impact of climate change on the GM and immune system in the development of illness may allow for the design of strategies aimed at reducing the harmful impact of climate and pollutants on health.

For instance, several adaptation and mitigation plans, including campaigns for behavioral changes and infrastructural modernization, aimed at lowering the emission of greenhouse gases, have been suggested to promote a sustainable and healthy future [127]. Environmental pollution and its negative effects on human health could be mitigated by further interventions, such as the promotion of public transportation, the enlargement of green spaces in urbanized contexts and the employment of accurate measuring stations [128]. Concerning GM, it would be beneficial to conduct extensive large-scale studies to discern probiotics to utilize for intervention. For instance, probiotics are being recommended as an effective and eco-friendly substitute for chemical-based preservatives. This movement has obtained significant attention from both the food industry and researchers, who are now actively working to create innovative food products containing probiotics [129]. The emergence of synbiotics represents a promising strategy for achieving functional foods aimed at modulating GM and delivering health benefits.

Future studies will need to combine technological and medical expertise to understand the specific probiotics to use and how to integrate them into foods, to enhance the quality of the gut microbiome and make it strongly functional for human health.

Being aware of the health consequences of climate change is an essential requirement to face the climate challenge.

## Figures and Tables

**Figure 1 diseases-12-00118-f001:**
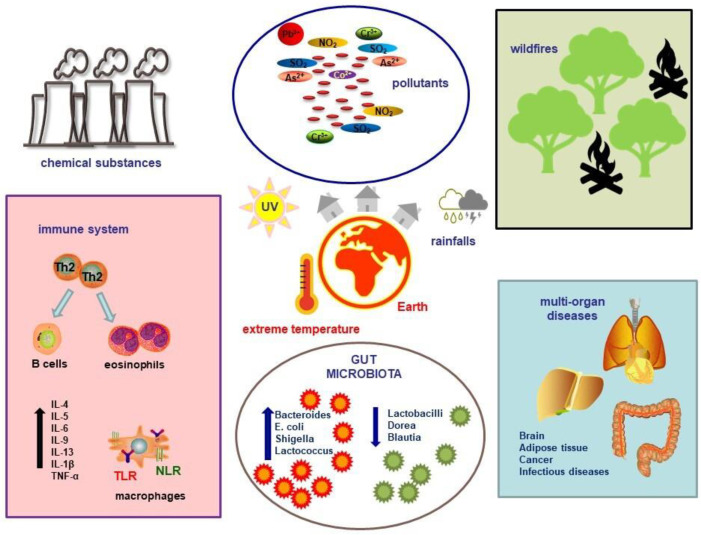
Climate change and pollution can influence the environment, leading to a reduction in environmental microbiome diversity and the production of inflammatory cytokines, which in turn contribute to the onset of multi-organ pathologies. Abbreviations: IL, interleukin; TNF, tumor necrosis factor; TLR, Toll-like receptor; NLR, NOD-like receptor; Th2, T helper 2; NO_2_, nitrogen dioxide; SO_2_, sulfur dioxide; Pb, lead; As, arsenic; Co, cobalt; Cr, chromium.

**Figure 2 diseases-12-00118-f002:**
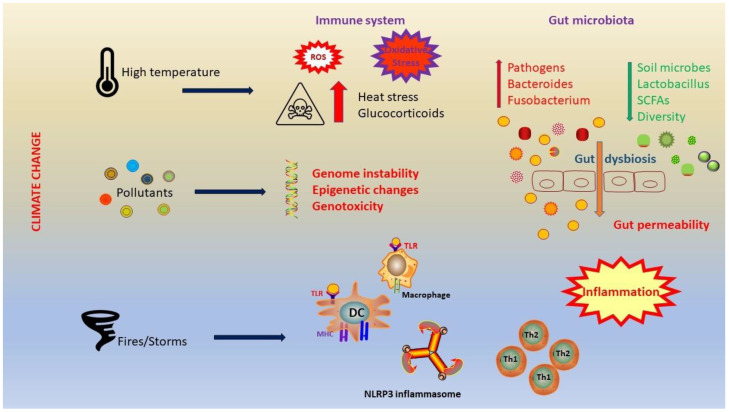
Climate change has a strong impact on the immune system and microbiota. Heat stress leads to neuroendocrine activation, with a consequent production of glucocorticoids. In addition, heat stress causes oxidative stress and increases intestinal permeability. A significant shift in microbiota composition occurs and it is characterized by increases in *Bacteroides* and *Fusobacterium*, and decreases in lactic acid bacteria. Dysbiosis leads to elevated gut permeability and inflammation. Abbreviations: ROS, reactive oxygen species; TLR, Toll-like receptor; MHC, major histocompatibility complex; DC, dendritic cell; NLRP3, nucleotide-binding domain, leucine-rich-containing family, pyrin domain-containing-3; Th1, T helper 1; Th2, T helper 2; SCFAs, short-chain fatty acids.

**Table 1 diseases-12-00118-t001:** Health consequences of climate change.

Climate-Related Event	Health Consequence	References
Global warming	Cardiovascular diseases Gastrointestinal infections Hepatitis and hepatocyte necrosis Cancer	[23,25,26,27,28]
Air pollutants (PM, greenhouse gases)	Exacerbation of chronic respiratory disorders and allergic diseases Cardio-cerebrovascular diseases Gastrointestinal inflammation Cancer	[17,19,22,29]
Extreme weather events (floods, hurricanes, fires)	Post-traumatic stress disorder, affective disorders and anxiety Reduced physical activity	[24,47,48]
Changes in rainfall patterns	Changes in the distribution of vectors Infectious diseases	[37]
Food contamination and impaired food availability	Gastrointestinal and systemic inflammation Metabolic diseases Immune imbalance Cancer	[30,31]

Abbreviations: PM, particulate matter.

**Table 2 diseases-12-00118-t002:** Alterations of GM related to temperature.

Temperature	Alteration of GM	References
High temperature	Declines in soil biodiversity	[9]
	Higher ratio Bacteroidetes-to-Firmicutes	[103]
	Increase in proportion of potentially pathogenic bacteria (*Escherichia*, *Shigella*, and *Lactococcus*)	[107]
	Increase of IL-10, IL-6, CXCL2, CCL4, CCL5, and CCL11	[107]
	Increase of ALT, ALP, creatinine, LDH, α-HBDH, and cholinesterase	[107]
	Decrease in probiotics (*Dorea*, *Blautia*, and *Lactobacillus*)	[107]
Cold temperature	Reduced richness and decreased α-diversity	[108]
	Increased gut permeability, apoptosis (cell death), inflammation, and oxidative stress	[109]
	Secretion of pro-inflammatory cytokines, such as IL-1β, IL-6, and TNF-α	[109]
	Secretion of butyric acid with the activation of the cAMP-PKA-pCREB signaling pathway	[110]
	SCFAs and LPS can modulate local levels of T3 increased by cold temperatures	[111]

Abbreviations: GM, gut microbiota; IL, interleukin; CXCL2, chemokine (C-X-C motif) ligand 2; CCL, chemokine (C-C motif) ligand; ALT, alanine transaminase; ALP, alkaline phosphatase; LDH, lactate dehydrogenase; α-HBDH, α-hydroxybutyrate dehydrogenase; TNF, tumor necrosis factor; SCFAs, short-chain fatty acids; LPS, lipopolysaccharides; T3, triiodothyronine.

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
