# Peer review of "The Impact of Climate Change on Immunity and Gut Microbiota in the Development of Disease"

_diseases, 2024, doi:10.3390/diseases12060118_

Round 1

Reviewer 1 Report

Comments and Suggestions for Authors

In the review article entitled The Impact of Climate Change on Immunity and Gut Microbiota in the Development of the Disease, the authors aimed to explore the influence of climate change on human health, focusing specifically on its effects on immune responses and gut microbiota.

The article is based on a good idea but has not been presented professionally.

The authors must revise the manuscript according to the following comments.

The authors must revise their manuscript according to the following comments.

Abstract

Should be revised and improvised.

Introduction

Extensive revision is required.

The world's weather is constantly shifting and increasingly unpredictable. Climate change was also shown to influence human health and both soil and gut microbiome.

But the authors haven’t mentioned a single line about it.

The authors must read the following articles to get a better idea and these articles are most recent and related. I am not co-author of any article of these articles.

1)    The interlink between climate changes, gut microbiota, and aging processes.

https://www.sciencedirect.com/science/article/pii/S2666149723000105

2)    Climate Change and Human Health: A Review of Allergies, Autoimmunity and the Microbiome

https://doi.org/10.3390/ijerph17134814

3)    Impact of climate change on immune responses and barrier defence

https://doi.org/10.1016/j.jaci.2024.01.016

2. Climate Change in Health and Disease

Climate change is also linked to the sustainability of Foods/Diets and various items.

As diets provide nourishment to various beneficial microbes present in the human body/ immune system.

The following articles must be read and cited.

https://doi.org/10.1080/19490976.2023.2297864 .

https://doi.org/10.3390/molecules28020491

https://doi.org/10.3390/nu15132956

The manuscript needs extensive revisions.

Figures should be added, only one figure is not enough.

The conclusion should be revised.

Tables should be provided only one table is not enough.

Comments on the Quality of English Language

In the review article entitled The Impact of Climate Change on Immunity and Gut Microbiota in the Development of the Disease, the authors aimed to explore the influence of climate change on human health, focusing specifically on its effects on immune responses and gut microbiota.

The article is based on a good idea but has not been presented professionally.

The authors must revise the manuscript according to the following comments.

The authors must revise their manuscript according to the following comments.

Abstract

Should be revised and improvised.

Introduction

Extensive revision is required.

The world's weather is constantly shifting and increasingly unpredictable. Climate change was also shown to influence human health and both soil and gut microbiome.

But the authors haven’t mentioned a single line about it.

The authors must read the following articles to get a better idea and these articles are most recent and related. I am not co-author of any article of these articles.

1)    The interlink between climate changes, gut microbiota, and aging processes.

https://www.sciencedirect.com/science/article/pii/S2666149723000105

2)    Climate Change and Human Health: A Review of Allergies, Autoimmunity and the Microbiome

https://doi.org/10.3390/ijerph17134814

3)    Impact of climate change on immune responses and barrier defence

https://doi.org/10.1016/j.jaci.2024.01.016

2. Climate Change in Health and Disease

Climate change is also linked to the sustainability of Foods/Diets and various items.

As diets provide nourishment to various beneficial microbes present in the human body/ immune system.

The following articles must be read and cited.

https://doi.org/10.1080/19490976.2023.2297864 .

https://doi.org/10.3390/molecules28020491

https://doi.org/10.3390/nu15132956

The manuscript needs extensive revisions.

Figures should be added, only one figure is not enough.

The conclusion should be revised.

Tables should be provided only one table is not enough.

Author Response

Rome, May 24th, 2024

Dear Editor of Diseases 

First of all, my coauthors and I would like to thank You sincerely for this opportunity of cooperation, allowing us to resubmit our paper (ID: diseases-2998591) after extensive revisions, for a possible publication upon “Diseases”.

We profoundly thank the reviewers for the comments and useful suggestions aimed at improving the paper.

We thank You for your constructive critique and we hope the review process has led to an improved manuscript.

If additional changes are warranted, we will make them.

We hope that this revised version of our manuscript may now be found suitable for publication.

Sincerely,

Rossella Cianci, MD, PhD

This is a point-by-point list of changes made in the paper:

REVIEWER 1

In the review article entitled The Impact of Climate Change on Immunity and Gut Microbiota in the Development of the Disease, the authors aimed to explore the influence of climate change on human health, focusing specifically on its effects on immune responses and gut microbiota.

The article is based on a good idea but has not been presented professionally.

The authors must revise the manuscript according to the following comments.

Abstract

Should be revised and improvised.

  • We have revised the abstract as requested.

Introduction

Extensive revision is required.

The world's weather is constantly shifting and increasingly unpredictable. Climate change was also shown to influence human health and both soil and gut microbiome.

But the authors haven’t mentioned a single line about it.

The authors must read the following articles to get a better idea and these articles are most recent and related. I am not co-author of any article of these articles.

1) The interlink between climate changes, gut microbiota, and aging processes. https://www.sciencedirect.com/science/article/pii/S2666149723000105

2) Climate Change and Human Health: A Review of Allergies, Autoimmunity and the Microbiome. https://doi.org/10.3390/ijerph17134814

3) Impact of climate change on immune responses and barrier defence. https://doi.org/10.1016/j.jaci.2024.01.016

-              We have extensively revised the Introduction as requested. We mentioned the impact of climate change on human health, soil and gut microbiome, that is further described in the following paragraphs. We cited the articles indicated.

  1. Climate Change in Health and Disease

Climate change is also linked to the sustainability of Foods/Diets and various items.

As diets provide nourishment to various beneficial microbes present in the human body/ immune system.

The following articles must be read and cited.

https://doi.org/10.1080/19490976.2023.2297864 .

https://doi.org/10.3390/molecules28020491

https://doi.org/10.3390/nu15132956

  • We have modified this paragraph as indicated. We discussed the link between climate change and food/diet and cited the articles indicated.

The manuscript needs extensive revisions.

  • We have made the changes as requested.

Figures should be added, only one figure is not enough.

  • We added a second figure as suggested.

The conclusion should be revised.

  • We have revised the conclusions.

Tables should be provided only one table is not enough.

  • We added a second table as suggested.

Reviewer 2 Report

Comments and Suggestions for Authors

The  research topic is interesting, contemporary and bring new findings to scientific community.  The authors done comprehensive research and analysis of the available literature about influence of climate changes on the environment and humans health. They done it by the investigations of the changes  in intestinal function based on behavior of the microorganism that live in digestive tract in addition authors investigate how those changes influence the behavior of the peoples and general health impact.

The comparison of the already published paper with similar research topic,  analysis of the published data and finding pattern of the behavior based on it.

The paper is well organized easy to read and understand and reader can easily made conclusions by reading it. The specific impact is in the presentation of the similar articles with pointing out most important finding in it.

The structure of the paper is excellent design and the presentation of the data makes it easy understandable. But I would like to suggest authors to put sub paragraphs i.e subtitles with specific research topic in cahpetr 4 and 5. It would make paper even more understandable.

The conclusion are supported by the presented data they summarize all important finding and give clear responses to the questions addressed in the design of the article.

The  number of the references is little bit high but in general every reference has its impact on the paper quality. 

The number of the figures and tables is adequate for the research and makes paper more understandable. the conclusions summarize all important premises and its supported by the discussion.

Comments on the Quality of English Language

the paper is well written, organized and easy to understand hence the English style and grammar are fine.

Author Response

Rome, May 24th, 2024

Dear Editor of Diseases 

First of all, my coauthors and I would like to thank You sincerely for this opportunity of cooperation, allowing us to resubmit our paper (ID: diseases-2998591) after extensive revisions, for a possible publication upon “Diseases”.

We profoundly thank the reviewers for the comments and useful suggestions aimed at improving the paper.

We thank You for your constructive critique and we hope the review process has led to an improved manuscript.

If additional changes are warranted, we will make them.

We hope that this revised version of our manuscript may now be found suitable for publication.

Sincerely,

Rossella Cianci, MD, PhD

This is a point-by-point list of changes made in the paper:

REVIEWER 2

The research topic is interesting, contemporary and bring new findings to scientific community.  The authors done comprehensive research and analysis of the available literature about influence of climate changes on the environment and humans health. They done it by the investigations of the changes  in intestinal function based on behavior of the microorganism that live in digestive tract in addition authors investigate how those changes influence the behavior of the peoples and general health impact.

The comparison of the already published paper with similar research topic, analysis of the published data and finding pattern of the behavior based on it. 

The paper is well organized easy to read and understand and reader can easily made conclusions by reading it. The specific impact is in the presentation of the similar articles with pointing out most important finding in it.

The structure of the paper is excellent design and the presentation of the data makes it easy understandable. But I would like to suggest authors to put sub paragraphs i.e subtitles with specific research topic in cahpetr 4 and 5. It would make paper even more understandable.

The conclusion are supported by the presented data they summarize all important finding and give clear responses to the questions addressed in the design of the article. 

The  number of the references is little bit high but in general every reference has its impact on the paper quality. 

The number of the figures and tables is adequate for the research and makes paper more understandable. the conclusions summarize all important premises and its supported by the discussion.

  • We thanks the reviewer. We have made the changes required.

Reviewer 3 Report

Comments and Suggestions for Authors

The manuscript summarized and discussed current research regarding changes in human immune responses and gut microbiota associated with climate change and offered a comprehensive view of the impact of climate change on human health and disease. This manuscript features excellent scientific writing, effectively presenting complex concepts in a clear and organized manner.

1. Line 3: Please remove “the” before Disease in the title.

2. Line 25: Please consider replacing “first” with “primary” or “predominant” to prevent ambiguity.

3. Please consider concluding the introduction by emphasizing the significance of the manuscript in advancing current knowledge of the impact of climate change on human health and disease.

4. Line 112: This sentence appears confusing. Please remove “and” and replace it with “in”.

5. Lines 134-135: This sentence appears confusing. Please correct the typo “wildl” and reconstruct the sentence to enhance clarity.

6. Line 169: "peculiar" might not be the most appropriate word choice. While "peculiar" can mean "particular" or "distinctive," it can also carry connotations of strangeness or oddity, which may not be the intended meaning here. Please consider replacing the world with an alternative, such as “specific”.

7. Figure 1 appears rather scattered and oversimplified. Please revise the figure to provide a clearer and more comprehensive illustration of the topic.

8. Lines 314-318: This sentence is rather long and convoluted. Please consider revising it to improve readability.

9. Please consider combining Table 1 and Table 2 to provide a more comprehensive presentation of the changes in GM to hot and cold temperatures.

10. Adding an extra figure that highlights the two primary sections covered—immune responses and gut microbiota—would significantly enhance the clarity and visual organization of this manuscript. Please include an additional figure to serve this purpose.

11. Generally, abbreviations should only be used if the term appears two or more times in the text. If a term only appears once in the manuscript, it's unnecessary to introduce them with abbreviations. Please conduct a thorough proofreading of the manuscript to remove any unnecessary abbreviations.

Comments on the Quality of English Language

The manuscript exhibits an excellent standard of English language usage, demonstrating a strong command of grammar, syntax, and clarity. While there are minor issues to be addressed, overall, the writing is highly proficient and effectively communicates the scientific content.

Author Response

Rome, May 24th, 2024

Dear Editor of Diseases 

First of all, my coauthors and I would like to thank You sincerely for this opportunity of cooperation, allowing us to resubmit our paper (ID: diseases-2998591) after extensive revisions, for a possible publication upon “Diseases”.

We profoundly thank the reviewers for the comments and useful suggestions aimed at improving the paper.

We thank You for your constructive critique and we hope the review process has led to an improved manuscript.

If additional changes are warranted, we will make them.

We hope that this revised version of our manuscript may now be found suitable for publication.

Sincerely,

Rossella Cianci, MD, PhD

This is a point-by-point list of changes made in the paper:

REVIEWER 3

The manuscript summarized and discussed current research regarding changes in human immune responses and gut microbiota associated with climate change and offered a comprehensive view of the impact of climate change on human health and disease. This manuscript features excellent scientific writing, effectively presenting complex concepts in a clear and organized manner.

  1. Line 3: Please remove “the” before Disease in the title.

- We remove it as requested.

  1. Line 25: Please consider replacing “first” with “primary” or “predominant” to prevent ambiguity.

- We have made the changes as requested.

  1. Please consider concluding the introduction by emphasizing the significance of the manuscript in advancing current knowledge of the impact of climate change on human health and disease.

- We modified the introduction as requested.

  1. Line 112: This sentence appears confusing. Please remove “and” and replace it with “in”.

- We have made the changes as requested.

  1. Lines 134-135: This sentence appears confusing. Please correct the typo “wildl” and reconstruct the sentence to enhance clarity.

- We have made the changes as requested.

  1. Line 169: "peculiar" might not be the most appropriate word choice. While "peculiar" can mean "particular" or "distinctive," it can also carry connotations of strangeness or oddity, which may not be the intended meaning here. Please consider replacing the world with an alternative, such as “specific”.

- We replaced the world with “specific”.

  1. Figure 1 appears rather scattered and oversimplified. Please revise the figure to provide a clearer and more comprehensive illustration of the topic.

- We modified the figure as requested.

  1. Lines 314-318: This sentence is rather long and convoluted. Please consider revising it to improve readability.

- We have made the changes as requested.

  1. Please consider combining Table 1 and Table 2 to provide a more comprehensive presentation of the changes in GM to hot and cold temperatures.

- We combined the indicated Tables as requested.

  1. Adding an extra figure that highlights the two primary sections covered—immune responses and gut microbiota—would significantly enhance the clarity and visual organization of this manuscript. Please include an additional figure to serve this purpose.

- We added an extra figure as requested.

  1. Generally, abbreviations should only be used if the term appears two or more times in the text. If a term only appears once in the manuscript, it's unnecessary to introduce them with abbreviations. Please conduct a thorough proofreading of the manuscript to remove any unnecessary abbreviations.

- We have made the changes as requested.

Round 2

Reviewer 1 Report

Comments and Suggestions for Authors

The authors have revised the manuscript and can be accepted for publication now.

Author Response

We thank the reviewer for her/his comment